# Sarcopenia Is an Independent Risk Factor for Subsequent Osteoporotic Vertebral Fractures Following Percutaneous Cement Augmentation in Elderly Patients

**DOI:** 10.3390/jcm11195778

**Published:** 2022-09-29

**Authors:** Shira Lidar, Khalil Salame, Michelle Chua, Morsi Khashan, Dror Ofir, Alon Grundstein, Uri Hochberg, Zvi Lidar, Gilad J. Regev

**Affiliations:** Department of Neurosurgery, Tel-Aviv Sourasky Medical Center, Sackler Faculty of Medicine, Tel Aviv University, Tel Aviv 6423906, Israel

**Keywords:** sarcopenia, osteoporosis, recurrent fractures, psoas, cross-sectional area

## Abstract

**Introduction**: Subsequent osteoporotic vertebral fractures (SOVF) are a serious complication of osteoporosis that can lead to spinal deformity, chronic pain and disability. Several risk factors have been previously identified for developing SOVF. However, there are conflicting reports regarding the association between sarcopenia and multiple vertebral compression fractures. As such, the goal of this study was to investigate whether sarcopenia is an independent risk factor of SOVF. **Methods:** This was a retrospective case–control study of elderly patients who underwent percutaneous vertebral augmentation (PVA) due to a new osteoporotic vertebral compression fracture (OVCF). Collected data included: age, sex, BMI, steroid treatment, fracture level and type, presence of kyphosis at the level of the fracture and bone mineral density (BMD). Identification of SVOFs was based on clinical notes and imaging corroborating the presence of a new fracture. Sarcopenia was measured using the normalized psoas muscle total cross-sectional area (nCSA) at the L4 level. **Results:** Eighty-nine patients that underwent PVA were followed for a minimum of 24 months. Average age was 80.2 ± 7.1 years; 58 were female (65.2%) and 31 male (34.8%). Psoas muscle nCSA was significantly associated with age (*p* = 0.031) but not with gender (*p* = 0.129), corticosteroid treatment (*p* = 0.349), local kyphosis (*p* = 0.715), or BMD (*p* = 0.724). Sarcopenia was significantly associated with SOVF (*p* = 0.039) after controlling for age and gender. **Conclusions:** Psoas muscle nCSA can be used as a standalone diagnostic tool of sarcopenia in patients undergoing PVA. In patients undergoing PVA for OVCF, sarcopenia is an independent risk factor for SOVF.

## 1. Introduction

Osteoporosis is a chronic bone disease characterized by a decrease in bone mass and increased risk of pathological fractures. Osteoporotic vertebral compression fractures (OVCF) are the most common complication of osteoporosis due to the high mechanical load of the spine [1]. Previous studies have found that the risk of fractures increases due to different causes such as the degree of osteopenia (loss of bone mass), presence of previous pathological fractures, and female gender [2,3].

As the population ages, osteoporotic fractures are becoming more common and identification of patients that are at increased risk of developing recurrent OVCF has gained importance. Osteoporotic vertebral compression fractures are an increasingly prevalent cause of intractable back pain, decreased mobility and prolonged bed riddance, carrying a significant socioeconomic burden [4].

Sarcopenia is a process of depletion of muscle mass in the body. Similar to osteoporosis, the incidence of this condition increases in geriatric populations and depends on various systemic conditions and diseases [5,6].

The diagnosis of sarcopenia is typically based on a combination of low handgrip strength, slow gait speed and decreased appendicular skeletal muscle mass in the upper and lower extremities as measured by dual X-ray absorptiometry (DEXA) [7,8]. Sarcopenia can also be estimated using measurement of paraspinal muscle cross sectional area (CSA) which closely correlates to the total body muscle mass [7,9,10,11,12]. Previous studies have used computed-tomography (CT) or magnetic resonance imaging (MRI) modalities to assess the total psoas CSA (tCSA) at the level of either the L3 or the L4 vertebra [13,14]. The psoas tCSA of patients can be normalized against the vertebral body area or body height. [12,15,16]. Sarcopenia can thus be estimated by low values of the normalized psoas CSA (nCSA) [16,17].

Previous studies have found that sarcopenia and frailty are correlated with increased complication rates, prolonged postoperative hospitalizations and an increase in overall mortality following major medical events such as oncological disease and major surgeries [16,17]. However, other studies did not find a clear association between poor clinical outcomes and sarcopenia following spine surgeries. [18]. Hida et al. reported that sarcopenia, measured as decreased leg skeletal mass index, was a risk factor for vertebral fractures in elderly Japanese women. In this study, the prevalence of sarcopenia was 42% and 25% in fracture and non-fracture groups, respectively [19]. On the contrary, Ashish et al. did not find that sarcopenia was an independent risk factor for recurrent fractures. However, they did find that sarcopenia was more prevalent in a subgroup of patients with previous OVCF [20]. A major complication of OVCF is subsequent osteoporotic vertebral fractures (SOVF) at adjacent levels. Development of multiple compression fractures can lead to spinal deformity, chronic pain and disability [2,21,22]. SOVFs after percutaneous vertebral augmentation are common, with an incidence of 12% to 52% of patients [23]. Therefore, the identification of risk factors may help in developing adjuvant interventions to improve clinical outcomes.

Refs. [5,6,7,8,9,10,11,12,13,14,15,16,17]. As such, the goal of this study was to investigate whether sarcopenia can be correlated to an increased risk of SOVFs following percutaneous vertebral augmentation.

## 2. Materials and Methods

### 2.1. Patient Population

This was a retrospective case–control study. The database of our tertiary hospital was searched for elderly patients who underwent percutaneous vertebral augmentation (PVA) due to VOFs of the thoracolumbar junction between the years 2007 and 2017 with a post-operative clinical follow-up of at least 24 months. Patients who developed SOVF and underwent a repeat single level PVA for their fracture formed the study group. Patients who did not suffer from SOVF following their first PVA formed the control group. Exclusion criteria included (1) age younger than 55 years, (2) high-energy trauma, (3) previous PVA before their enrollment to this study, (4) spinal deformities due to previous VCFs (i.e., local kyphosis), (5) pathological fractures (due to malignancy or infection), and (6) lack of pre-operative CT or clinical follow-up. Data collected included: age, sex, BMI, corticosteroid treatment, fracture level, presence of kyphosis at the level of the fracture and degree of osteoporosis as measured using mean Hounsfield units (HU) values for L4 and L5 vertebrae. Identification of SVOFs was based on clinical notes and imaging corroborating the presence of a new fracture (X-rays, CT, magnetic resonance imaging or bone scintigraphy).

The study received institutional review board approval and informed consent was waived due to its retrospective nature.

### 2.2. Radiological Analysis

Subsequent osteoporotic vertebral fractures were assessed on preoperative thoracolumbar CT scans according to AO Spine thoracolumbar injury classification system [24]. Preoperative sagittal plane computer tomography (CT) images were used to measure local kyphosis at the level of the fracture by measuring the angle between the upper-endplate of the caudal vertebra and the lower-endplate of the superior vertebra adjacent to the facture. Bone mineral density (BMD) on CT has been described as a tool for assessment of the risk of biomechanical complications in spinal surgery [25]. CT machines with automatic exposure control allow for easy measurement of tissue density (its linear attenuation coefficient), expressed by Hounsfield units (HU). Similar to the technique described by Schreiber et al. using HU in the cancellous bone of the vertebral body [26]. The images were analyzed using Centricity Universal Viewer, Next-gen Picture Archiving and Communications System (PACS), (GE Healthcare, Wood Dale, IL, USA). HU measurements were performed for all vertebrae included in the CT scan, at the L4 and L5 levels, using 5 mm elliptical range of interest (ROIs). The ROIs defined were restricted to cancellous bone and avoided obvious bony lesions (such as hemangiomas) and the posterior venous plexus. Average HU values were than calculated from three ROI measurements per patient in the axial plane.

Psoas measurements were obtained from axial CT images using Image J software, (U.S. National Institutes of Health, Bethesda, MD, USA) [9]. Muscles were measured bilaterally at the midlevel of the L4–5 facet. Psoas muscle tCSA was calculated as the sum of the right and left psoas muscles CSA. Vertebral body CSA was measured at the inferior endplate of L4. Psoas nCSA was then calculated as the ratio of bilateral psoas CSA to vertebral body CSA to normalize for body habitus [12,14,15,27]. Sarcopenia was defined as psoas nCSA less than the sex specific median in accordance with previous studies [15,16,28].

### 2.3. Statistical Methods

Statistical analysis was performed using R version 3.6.3 (R Foundation for Statistical Computing, Vienna, Austria) (http://www.r-project.org, accessed on 10 January 2022). The dependent variable is the occurrence of subsequent osteoporotic vertebral fracture. The independent variables were age, gender, fracture classification, primary PVA procedure, use of corticosteroids, local kyphosis, BMD, and psoas CSA. In univariable analysis, variables were compared between groups by Fisher’s exact test for categorical variables and the Wilcoxon signed-rank test for numerical variables. Statistical significance was defined as *p* < 0.05. Multivariable analysis was performed using multiple logistic regression. Parameters were included in multivariable analysis based on clinical significance and statistical significance in univariable analysis.

## 3. Results

In total, 287 patients were initially identified, and 89 suitable patients remained following application of the exclusion criteria. Mean age was 80.2 ± 7.1 years; 58 were female (65.2%) and 31 male (34.8%). Patient demographics and clinical variables are presented in Table 1.

Patients’ post-operative follow-up time following the first PVA ranged between 1 and 79 months. During this period, SOVFs were diagnosed in 26 (29.2%) of the patients. Sarcopenia was diagnosed in 65.4% (17) of patients with SOVF compared to 38.1% (24) of the non-sarcopenic patients. (*p* = 0.5)

In patients diagnosed with SOVF, both psoas tCSA and nCSA were lower compared to the control group. Although both parameters did not reach statistical significance, this difference was much more prominent for the nCSA (*p* = 0.06) compared to the tCSA (*p* = 0.6). No significant differences were found between the groups when comparing age, fracture classification, PVA technique, local kyphosis at the level of the fracture or chronic steroid use. Table 2

Psoas tCSA was significantly associated with gender (*p* = 0.017) but not with age (*p* = 0.216), use of corticosteroids (*p* = 0.685), local kyphosis (*p* = 0.219), or BMD (*p* = 0.420), whereas Psoas nCSA was significantly associated with age (*p* = 0.031) but not with gender (*p* = 0.129), use of corticosteroids (*p* = 0.349), local kyphosis (*p* = 0.715), or BMD (*p* = 0.724).

In multivariable analysis, sarcopenia was significantly associated with SOVF (*p* = 0.039) after controlling for age and gender. No other independent predictors of adjacent level fracture were identified (see Table 3).

## 4. Discussion

Our study establishes sarcopenia as an independent risk factor for subsequent vertebral fragility fractures in individuals who had previous PVA for a first osteoporotic fracture. Furthermore, we found that low psoas muscle nCSA can be used as a standalone diagnostic tool for risk assessment of SOVF in patients undergoing PVA. The associations found between the psoas muscle CSA, older age and female gender reflect the higher prevalence of sarcopenia in these subgroups of patients, which is similar to findings of previous studies [27,29,30].

Several possible mechanisms can be proposed to explain why sarcopenia increases the risk of SVOF. The first is that bone and muscle interconnect not only because of their adjacent surfaces but due to their chemical and metabolic properties [30]. As such, sarcopenic patients may have relatively weaker bones that are more susceptible to fracture compared to non-sarcopenic patients. Additionally, weakness of the paraspinal and core muscles may decrease the ability of the spinal column to manage external loads of daily activities, which in turn exposes it to repeated fractures.

Previous studies have found that low bone density and alterations in the spinal sagittal balance due to an increase in kyphosis are independent risk factors for SOVF. Thus, the main strategies for preventing SOVF to date have included medications to improve bone density, correction of spinal deformity and improvement of the surgical techniques for PVA. In a recent meta-analysis, female gender, lower T-score, thoracolumbar junction fracture, intravertebral cleft, higher injected cement volume, and intradiscal cement leakage were identified as independent risk factors for adjacent level fracture, whereas BMI, use of corticosteroids and Cobb angle change were not [31]. Similarly, we did not find an association between postoperative Cobb angle or the use of corticosteroids with SOVF. Female gender was more prevalent in the SOVF group compared to the control group, nearly reaching statistical significance (*p* = 0.054). In contrast, the degree of osteopenia, as measured on CT scans, was not associated with an increased risk of SOVF. This may be explained by the fact that our measurements were not normalized for large populations, as is the case with DEXA scans measurements, in order to calculate the T-score value.

Wang et al. have previously studied the association between sarcopenia and SOVF [27]. Similar to our results, they too found that sarcopenia was an independent risk factor of SOVF in their patient cohort. This risk was further associated with lower BMD, advanced age and female sex. These authors used a patients cohort consisted solely of Chinese ethnicity who similarly underwent PVA for an osteoporotic vertebral fracture. Although the rate of SVOF in both cohorts was similar to our results (29% vs. 27%), the diagnosis of sarcopenia was two-fold more prevalent in our study population (65.4% vs. 32.8%).

This difference may be attributed to several factors, the most important of which is the method chosen for the assessment of sarcopenia. In our study, we used a simple and approachable measurement for the diagnosis of sarcopenia focusing only on the psoas muscle and adjacent vertebral body CSA, whereas Wang et al. utilized much more complex and cumbersome diagnostic criteria that used measurements of both the psoas and the posterior paraspinal muscles CSA coupled with functional parameters measured by grip strength and gait speed.

Age and sex distribution between the cohorts are additional factors that may have contributed to the large discrepancy in the prevalence of sarcopenia between the two studies. Our patients were a decade older than the patients included in Wang’s study (80.2 ± 7.1 vs. 70.61 ± 8.87). Additionally, in Wang’s study, the sarcopenic cohort consisted of a vast majority of females compared to the percentage of female patients in our cohort (85% vs. 65%). Lastly, the population ethnicities included in each study were different. While our population is Middle Eastern/Caucasian, Wang at al. included patients of Asian origin, who are renowned for having low bone mineral density, which may have contributed to the recurrent osteoporotic vertebral fractures more than the sarcopenia [32,33]. It may therefore be deduced that the addition of functional assessment to the imaging-based diagnosis of sarcopenia would reduce the sensitivity and enhance the specificity of the diagnosis. However, we were able to establish that, even with the use of a simpler and straightforward diagnostic tool, a clinically relevant diagnosis of sarcopenia can be achieved. This significantly increases the useability for routine preoperative assessment of patients treated for OVCF.

There are some limitations to this study. First this is a retrospective study in which some additional factors that could influence the risks for SOVF, such as body mass index (BMI), other medical conditions and routine medical treatments, were unavailable. Although we based our measurement techniques for diagnosing sarcopenia and osteoporosis on previous studies, there is still disagreement in the literature regarding the optimal method and cutoff values for the diagnosis of these conditions. As such, comparing our results to previous studies may be misleading. Lastly, our patient cohort was composed of a relatively small number of patients of Caucasian ethnicity.

Further research is required in order to validate our results with larger patient cohorts and different ethnic groups, as well as to investigate whether specific treatment strategies aimed to increase body lean mass can additionally reduce the risk of recurrent osteoporotic fractures.

## 5. Conclusions

The result of our study further solidified the role of sarcopenia as a risk factor for recurring fracture in osteoporotic patients. Clinical estimation of this risk assessment could be reached using a straightforward analysis of the patients’ psoas muscle nCSA, without the need for additional functional tests. We recommend that this technique be used routinely in the preoperative evaluation of patients diagnosed with OVCF.

## Figures and Tables

**Table 1 jcm-11-05778-t001:** Patient demographics and baseline characteristics.

	No Adjacent Level Fracture	Adjacent Level Fracture	*p*
	N = 63	N = 26	
Age (years)	80.2 ± 7.2	80.1 ± 7.1	0.986
Male	26 (41.3%)	5 (19.2%)	0.054
Female	37 (58.7%)	21 (80.8%)	
Use of corticosteroids	3 (4.8%)	2 (8.0%)	0.62

**Table 2 jcm-11-05778-t002:** Univariable analysis for associations between patient characteristics and adjacent level fracture.

	No Adjacent Level Fracture N = 63	Adjacent Level Fracture N = 26	*p*
PVA Procedure			
Kyphoplasty	38	20	
Vertboplasty	9	4	0.99
Not Available	16	2	
AO Spine Thoracolumbar Injury Classification System			
A0	10	2	
A1	27	12	
A2	2	1	0.51
A3	8	4	
Not Available	0	1	
Psoas tCSA (cm^2^) Psoas nCSA	8.3 ± 2.5 0.75 ± 0.20	7.6 ± 1.8 0.67 ± 0.14	0.435 0.064
Psoas nCSA < Q2 Psoas nCSA ≥ Q2	24 (38.1%) 39 (61.9%)	17 (65.4%) 9 (34.6%)	0.054
Local kyphosis (degrees)	8.6 (2.0–15.6)	6.4 (−2.0–16.5)	0.642
BMD (HU)	71.9 ± 35.7	71.0 ± 41.0	0.875

PVA: percutaneous vertebral augmentation; BMD: bone mineral density; tCSA: total cross-sectional area; nCSA: normalized cross sectional area.

**Table 3 jcm-11-05778-t003:** Multivariable analysis for independent predictors of adjacent level fracture.

	Predictor	OR (95% CI)	*p*
Adjacent level fracture	Psoas nCSA < Q2	2.79 (1.05–7.41)	0.039
	Age	1.00 (0.93–1.07)	0.986
	Male gender	0.35 (0.11–1.08)	0.068

nCSA normalized cross sectional area.

## Data Availability

Not applicable.

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
