# Peer review of "Sarcopenia Is an Independent Risk Factor for Subsequent Osteoporotic Vertebral Fractures Following Percutaneous Cement Augmentation in Elderly Patients"

_jcm, 2022, doi:10.3390/jcm11195778_

Round 1

Reviewer 1 Report

The authors should be congratulated for conducting a research study on a relevant topic.

However, there are significant issues in the methodology that must be addressed before this manuscript can be considered for publication. 

M&M

- lines 92-94: "average values of HU of the vertebral body were measured at the L4 and L5 levels using 5 mm elliptical ROIs in the horizontal plane". Please specify how many measurements were made in each level to calculate the average values of HU

- lines 98-99: "Psoas nCSA was then calculated as the ratio of bilateral psoas CSA to vertebral body CSA to normalize for body habitus[16]". Please provide the rationale and reference for this normalization, since the reference provided (doi: 10.1007/s00198-019-05144-x) does not support this normalization.

- lines 99-101: " Sarcopenia was defined as psoas nCSA less than the sex specific median in accordance with previous studies [17]". The reference (doi: 10.1007/s00586-010-1479-6) does not mention psoas nCSA.

Results

- Table 1: Adjacent level fracture is not a baseline characteristics. The Table is badly-formatted and it is not clearly which values refer to vertebroplasty and kyphoplasty. Please specify the meaning of 'AO Class' and include the classification.

- lines 116-117: "Sarcopenia defined as psoas rCSA less than the sex specific median was diagnosed..." Please provide values and calculation method for these sex-specific medians

- lines 117 and following: what does rCSA mean?

- Why only age and gender were included in multivariate analysis? 

- Why was not the surgical technique (vertebroplasty versus kyphoplasty) included in the univariable and multivariable analyses?

Discussion:

- lines 153-155: "Furthermore, we found that the diagnosis of sarcopenia using lumbar spine CT scans can be used as a stand-alone diagnostic tool for risk assessment of SOVF in patients undergoing PVA". The data do not support that CT scans can be used to diagnose sarcopenia.

Conclusion:

- lines 221-223: "Clinically relevant diagnosis of sarcopenia for this risk assessment could be reached using a straight forward analysis of the patients lumbar spine CT scan without the need for additional functional tests". Please see my previous comment anf rephrase.

Author Response

Reviewer 1

Comments and Suggestions for Authors

The authors should be congratulated for conducting a research study on a relevant topic.

Thank you,

However, there are significant issues in the methodology that must be addressed before this manuscript can be considered for publication.

M&M

- lines 92-94: "average values of HU of the vertebral body were measured at the L4 and L5 levels using 5 mm elliptical ROIs in the horizontal plane". Please specify how many measurements were made in each level to calculate the average values of HU.

Average HU values were measured on 3 elliptical ROIs in the axial plane for each patients. We added  a detailed descriptions of the technique was added to the text (p. 4 line 6-10).

- lines 98-99: "Psoas nCSA was then calculated as the ratio of bilateral psoas CSA to vertebral body CSA to normalize for body habitus[16]". Please provide the rationale and reference for this normalization, since the reference provided (doi: 10.1007/s00198-019-05144-x) does not support this normalization.

We have corrected and added references [12, 14, 15, 28] of studies that previously used this method.

- lines 99-101: " Sarcopenia was defined as psoas nCSA less than the sex specific median in accordance with previous studies [17]". The reference (doi: 10.1007/s00586-010-1479-6) does not mention psoas nCSA.

We have corrected and added references [15, 16,29].

Results

- Table 1: Adjacent level fracture is not a baseline characteristics. The Table is badly-formatted and it is not clearly which values refer to vertebroplasty and kyphoplasty. Please specify the meaning of 'AO Class' and include the classification.

We have re-formatted table 1.  AO Spine thoracolumbar injury classification system was used for fracture classification and we  added reference [25]  

- lines 116-117: "Sarcopenia defined as psoas rCSA less than the sex specific median was diagnosed..." Please provide values and calculation method for these sex-specific medians

We apologize for this mistake, we are referring to nCSA and not rCSA. We have provided the references and calculation method In the methods section (p.4 line 16-18) 

- lines 117 and following: what does rCSA mean?

Please see previous reply.

- Why only age and gender were included in multivariate analysis?

Parameters included in multivariable analysis were based on statistical significance in the univariable analysis. Surgical technique, use of corticosteroids, local kyphosis, and BMD were not significantly associated in univariable analysis with subsequent osteoporotic vertebral fracture, and therefore were not included in the multivariable analysis. 

- Why was not the surgical technique (vertebroplasty versus kyphoplasty) included in the univariable and multivariable analyses?

We added the surgical technique and fracture classification to the list of variables included in the  univariable analysis. Since both variables were not significantly associated with SOVF in univariable analysis  they were not included in the multivariable analysis. 

Discussion:

- lines 153-155: "Furthermore, we found that the diagnosis of sarcopenia using lumbar spine CT scans can be used as a stand-alone diagnostic tool for risk assessment of SOVF in patients undergoing PVA". The data do not support that CT scans can be used to diagnose sarcopenia.

We have rephrased this  sentence to :” we found that low psoas muscle nCSA can be used as a stand-alone diagnostic tool for risk assessment  of SOVF in patients undergoing PVA.”

Conclusion:

- lines 221-223: "Clinically relevant diagnosis of sarcopenia for this risk assessment could be reached using a straight forward analysis of the patients lumbar spine CT scan without the need for additional functional tests". Please see my previous comment anf rephrase

We have rephrased this  sentence to : “Clinical estimation  of  this risk assessment could be reached using a straight forward analysis of the patients’ Psoas nCSA”

Reviewer 2 Report

General: This was a concise manuscript providing data for the association between sarcopenia and SOVF. The data will provide a useful reference source in developing interventions to improve clinical outcomes. The authors have included some limitations of the study, but the discussion section is an over interpretation of the results of the study, particularly as other factors influencing SVOA were not explored. The authors have also not explained the specific findings in the manuscript which have led to some of the conclusions in the discussion section.

Specific comments

Abstract

“Several risk factors have been previously identified as risk factors for developing SOVF” – sentence needs to be paraphrased

Introduction

“Previous studies have found that the risk of fractures increases due to different causes 35 such as the degree of osteopenia (loss of bone mass) , presence of previous pathological 36 fractures, lower body mass index (BMI) and female gender” This needs reference(s)

Line 37: …” lower body mass index (BMI) and female gender” –there is conflicting evidence around the association between lower body mass index and vertebral fractures. Please paraphrase.

Methods

Why exclude those younger than 55?

Statistical methods

What was considered independent/dependent variable? Please state these in this section

Discussion

The authors should provide an explanation/speculation of the associations between age/gender, and Psoas tCSA abd Psoas nCSA.

I believe paragraphs 1 and 2 (lines 135-150) should be in the introductory session, not discussion.

Lines 153- 155 “Furthermore, we found that the diagnosis of sarcopenia using lumbar spine CT scans 153 can be used as a stand-alone diagnostic tool for risk assessment of SOVF in patients undergoing PVA” how do the results of this study explain this?

Line 172 “Female gender was more prominent in the SOVF group compared to the control group 172 and nearly reaching statistical significance (p=0.5)” This is not exactly “nearly” reaching statistical significance.

T-sore value (lines 176) typo.

Lines 197- “While our population is middle eastern/Caucasian Wang at al. included patients of Asian origin 198 who are renown for having low bone mineral density.” is there a reference for this? Reference 23 does not include this.

Author Response

Comments and Suggestions for Authors

General: This was a concise manuscript providing data for the association between sarcopenia and SOVF. The data will provide a useful reference source in developing interventions to improve clinical outcomes.

Thank you,

The authors have included some limitations of the study, but the discussion section is an over interpretation of the results of the study, particularly as other factors influencing SVOA were not explored. The authors have also not explained the specific findings in the manuscript which have led to some of the conclusions in the discussion section.

We have made extensive revisions to the manuscript according to your specific recommendations which we believe  significantly improved the clarity and accuracy of the discussion and conclusion sections.

Specific comments:

Abstract

“Several risk factors have been previously identified as risk factors for developing SOVF” – sentence needs to be paraphrased

We have rephrased this  sentence

Introduction –

“Previous studies have found that the risk of fractures increases due to different causes 35 such as the degree of osteopenia (loss of bone mass) , presence of previous pathological 36 fractures, lower body mass index (BMI) and female gender” This needs reference(s)

We have added references [2,3].

Line 37: …” lower body mass index (BMI) and female gender” –there is conflicting evidence around the association between lower body mass index and vertebral fractures. Please paraphrase.

We have rephrased this  sentence and removed the low BMI mention.

Methods

Why exclude those younger than 55?

We  have focused this study on elderly medical patients. As such we excluded patients younger than 55. We have revised the title and text to reflect this.

Statistical methods

What was considered independent/dependent variable? Please state these in this section

The dependent variable is occurrence of subsequent osteoporotic vertebral fracture. The independent variables Included:  age, gender, surgical technique , fracture classification, use of corticosteroids, local kyphosis, BMD, and Psoas CSA.  We have added this statement to the method section

Discussion

The authors should provide an explanation/speculation of the associations between age/gender, and Psoas tCSA abd Psoas nCSA.

We  have Added an explanation to this association (Page 6 line 9-11).

I believe paragraphs 1 and 2 (lines 135-150) should be in the introductory session, not discussion.

We have moved this paragraph to the introduction.

Lines 153- 155 “Furthermore, we found that the diagnosis of sarcopenia using lumbar spine CT scans 153 can be used as a stand-alone diagnostic tool for risk assessment of SOVF in patients undergoing PVA” how do the results of this study explain this?

We have rephrased this  sentence to :” we found that low psoas muscle nCSA can be used as a stand-alone diagnostic tool for risk assessment  of SOVF in patients undergoing PVA.”

Line 172 “Female gender was more prominent in the SOVF group compared to the control group 172 and nearly reaching statistical significance (p=0.5)” This is not exactly “nearly” reaching statistical significance.

We have corrected this typo to (p=0.054).

T-sore value (lines 176) typo

Corrected.

Lines 197- “While our population is middle eastern/Caucasian Wang at al. included patients of Asian origin 198 who are renown for having low bone mineral density.” is there a reference for this? Reference 23 does not include this  

We have added a reference [34] to support this statement.